# Joint Probabilities Approach to Quantum Games with Noise

**DOI:** 10.3390/e25081222

**Published:** 2023-08-16

**Authors:** Alexis R. Legón, Ernesto Medina

**Affiliations:** 1Departamento de Física, Universidad Técnica Federico Santa María, Av. España 1680, Casilla 110 V, Valparaíso 2390123, Chile; 2Laboratorio de Física Estadística de Medios Desordenados, Instituto Venezolano de Investigaciones Científicas (IVIC) Carretera Panamericana, Km 11, Altos de Pipe, Caracas 1020A, Venezuela; 3Departamento de Física, Colegio de Ciencias e Ingeniería, Universidad San Francisco de Quito, Diego de Robles y Vía Interoceánica, Quito 170901, Ecuador

**Keywords:** quantum games, quantum strategies, noise, correlation measure, dilemma dissolution

## Abstract

A joint probability formalism for quantum games with noise is proposed, inspired by the formalism of non-factorizable probabilities that connects the joint probabilities to quantum games with noise. Using this connection, we show that the joint probabilities are non-factorizable; thus, noise does not generically destroy entanglement. This formalism was applied to the Prisoner’s Dilemma, the Chicken Game, and the Battle of the Sexes, where noise is coupled through a single parameter μ. We find that for all the games except for the Battle of the Sexes, the Nash inequalities are maintained up to a threshold value of the noise. Beyond the threshold value, the inequalities no longer hold for quantum and classical strategies. For the Battle of the sexes, the Nash inequalities always hold, no matter the noise strength. This is due to the symmetry and anti-symmetry of the parameters that determine the joint probabilities for that game. Finally, we propose a new correlation measure for the games with classical and quantum strategies, where we obtain that the incorporation of noise, when we have quantum strategies, does not affect entanglement, but classical strategies result in behavior that approximates quantum games with quantum strategies without the need to saturate the system with the maximum value of noise. In this manner, these correlations can be understood as entanglement for our game approach.

## 1. Introduction

The study of information processing has led to the birth of the quantum equivalent of classical games or quantum games (QG) [1,2,3,4,5]. While in classical games (CG), cooperation among players has expanded the possibilities for game outcomes and equilibria, QG has expanded these possibilities even further. The latter works have shown that quantum strategies can result in responses that often challenge our classical intuition. In this sense, entanglement allows the *dilemma* of CG to be broken [2,3,6]. Quantum games provide mathematical configurations to explore competitive interactions between quantum and classical players [1,7], in emblematic CG, such as the Prisoner’s Dilemma [2,3], the Chicken Game [3] and the Battle of the Sexes [8]. Also, there are important implications to quantum multiplayer games [9,10,11,12] interactions in the nascent field of quantum networks [13].

Quantum games are also subject to the effects of noise, which introduces mathematical operations that significantly modify the initial and final quantum state of the game. The incorporation of noise in QG has been spearheaded by Johnson [14]. Quantum games with noise must account for the influence of decoherence, and assessing the robustness against decoherence, which is always present, is a central issue in any practical implementation. Decoherence can take many forms, including dephasing (unitary) and dissipation (non-unitary) effects on quantum states. For this, we will determine the quantum game with noise in terms of Eisert’s [2,3] formulation of quantum games.

The consideration of joint probabilities in quantum games has been conceived in various works [7,15,16,17,18]. The joint probabilities can characterize games in a very precise way; if these are factorizable, it will be a classical game, and if they are not, we will have a quantum game [17]. From the work of Iqbal, Chappell, and Abbott [18], the pay-off relationships of the players are written in such a way that it allows us to obtain a non-factorizable game directly from the factorizable game, defining a function of the players’ strategies that satisfies certain restrictions. Therefore, canonical quantum game schemes can be described from the joint probabilities. This is advantageous since elements of the quantum information theory can be combined with the non-factorizable joint probabilities.

The full connection between the joint probabilities with Eisert quantum games has recently been achieved [6]. However, the connection between the joint probabilities approach and noise in the Eisert formulation has not been realized so far. In this work, we establish that connection by describing the quantum game with noise from the joint probabilities involved in the pay-off, inspired by the formalism proposed by [18]. To this end, we write the parameters $\epsilon_{i}$, proposed by [6,18], that correspond to the scheme of joint probabilities in terms of the joint probabilities with noise into the quantum games (Eisert’s quantum games). Furthermore, we determine the inequalities for the Nash Equilibrium (NE) for quantum games with noise and see the applications of this connection to three of the canonical games in game theory.

A necessary problem to address in quantum information theory is entanglement [19,20,21]. Here, we further develop the treatment of entanglement in the context of quantum games noise within the joint probabilities approach. A first approach to the problem without noise was advanced in reference [6], and this is the point of departure to include the effects of noise. In this sense, we review our previous proposal for a new correlation measure from joint probabilities in quantum games. In this way, we can quantify the entanglement in our game approach through a correlation measure and subsequently determine for noisy quantum games the effect of entanglement.

This work is divided as follows: first, we will start from the approach of [22] in order to write Eisert’s quantum games with noise [2,3]. Following that, starting from the approach of [18], a scheme of joint probabilities is proposed for Eisert’s quantum games with noise. Then we build the connection of the joint probabilities for the Eisert quantum game with noise, the inequalities associated with the NE, and the restrictions corresponding to the Prisoner’s Dilemma, the Chicken Game, and the Battle of the Sexes. The symmetry of joint probabilities in quantum games with noise is analyzed, differentiating the Battle of the Sexes from all the other reference games analyzed. Finally, we propose a correlation measure for quantum games with noise from their joint probabilities.

## 2. Eisert Quantum Games with Noise

Regardless of the particular model used, e.g., a quantum channel with noise, or a quantum game with decoherence [22,23,24,25], the model can be described as follows:(1)$$\begin{array}{ccc} \rho_{i}=\rho_{0}=|\psi_{0}\rangle \langle \psi_{0}| & \; & \mathrm{Initial}\mathrm{state}\\ \rho_{1}=\hat{J}\rho_{0}{\hat{J}}^{†} & \; & \mathrm{Entanglement}\\ \rho_{2}=D\left(\rho_{1},\mu_{1}\right) & \; & \mathrm{Partial}\mathrm{decoherence}\\ \rho_{3}=\left({⊗}_{k=1}^{N}{\hat{M}}_{k}\right)\rho_{2}{\left({⊗}_{k=1}^{N}{\hat{M}}_{k}\right)}^{†} & \; & {\mathrm{Player}}^{\prime }s\mathrm{movements}\\ \rho_{4}=D\left(\rho_{3},\mu_{2}\right) & \; & \mathrm{Partial}\mathrm{decoherence}\\ \rho_{5}={\hat{J}}^{†}\rho_{4}\hat{J} & \; & \mathrm{Dis}\text{-}\mathrm{entanglement}\\ \rho_{f}=\rho_{5} & \; & \mathrm{Final}\mathrm{state} \end{array}$$
where $|\psi_{0}\rangle =|00\ldots 0\rangle$ represents the initial state of the *N* qubits. The D(ρ,μ) function is a completely positive map that generates decoherence to the ρ state controlled by the probability μ. $\hat{J}$ is an operator that entangles while ${\hat{J}}^{†}$ disentangles the players qubits. ${\hat{M}}_{k}$ for k=1,…,N, represents the move of the player *k* associated with the operators $\pi_{i}$, that are the Kraus operators associated with the possible strategies of the game [2,3].

The expectation value of the pay-off of the *k*-th player $\Pi_{k}$, is computed as
(2)$$\Pi_{k}=\sum_{\alpha }\pi_{\alpha }\rho_{f}\pi_{\alpha }^{†}\eta_{\alpha }^{k},$$
where $\eta_{\alpha }^{k}$ represents the payment to a player of a game outcome and is a numerical value of the desirability of that outcome for the player [4], this player is associated with *k*. A measurement is made on the computational basis on $|\psi_{f}\rangle$, and the pay-offs are determined using the pay-off matrix gives numerical values to the players’ pay-offs for all the game outcomes of the classical game [4] (see [6] for the formal definition). The two classical pure strategies are the identity and the bit flit operator (see [26]). The classical game becomes a subset of the quantum game by requiring *J* to commute with the direct product of *N* classical moves. Games with more than two pure classical strategies associate each player with a number of qubits. The expression (Equation 2) represents an extension of the quantum game protocol proposed by Eisert [2,3], by introducing a quantum operation associated with noise, understood as decoherence in the protocol in (Equation 1), that is, the pay-off is (Equation 2) continues to represent the same pay-off for the quantum game in general, only now for a quantum game with noise.

The set of quantum strategies for this type of quantum game is ${𝒮}^{\left(𝒮\right)}=\left\{U\left(\theta ,\phi ,\beta \right):0\leq \theta \leq \pi ,0\leq \;\phi ,\beta \leq \pi /2\right\}$, where
(3)$$U\left(\theta ,\phi ,\beta \right)=\left(\begin{array}{cc} e^{i\phi }\cos\frac{\theta }{2} & -ie^{i\beta }\sin\frac{\theta }{2}\\ -ie^{-i\beta }\sin\frac{\theta }{2} & e^{-i\phi }\cos\frac{\theta }{2} \end{array}\right),$$
is a super-operator in SU(2). ${𝒮}^{\left(𝒮\right)}$ is called a superset of strategies. The *k*-th player’s move is $U(\theta_{k},\phi_{k},\beta_{k})$. There is some arbitrariness in the representation of operators. Nevertheless, different representations only lead to changes in a global phase in the final state.

After choosing the operation to represent the function D(ρ,μ) in (Equation 1), that is, the quantum channel, we take the quantum phase damping channel as in [22]. We are then able to quantify the consequences of decoherence in a bipartite quantum game. The unitary operators of the quantum game are represented by
(4)$$U\left(\theta ,\phi \right)=\left(\begin{array}{cc} e^{i\phi }\cos\frac{\theta }{2} & \sin\frac{\theta }{2}\\ -\sin\frac{\theta }{2} & e^{-i\phi }\cos\frac{\theta }{2} \end{array}\right).$$
with θ∈[0,π] and ϕ∈[0,π/2]. This determines the strategies for the Eisert quantum game [2,3] for one (sets ϕ=0) and two-parameter sets of strategies, respectively. Then, we evaluate $\beta_{A}=\beta_{B}=\frac{\pi }{2}$ into (Equation 64) (see Appendix B), and use trigonometric identities sin(x+y)−sin(x−y)=2sin(y)cos(x) and sin(x+y)+sin(x−y)=2sin(x)cos(y), so we have
(5)$$\begin{array}{c} \Pi_{k}\left(\theta_{A},\phi_{A},\theta_{B},\phi_{B},\mu \right)=\eta_{1}^{k}\left\{\frac{1}{2}c_{A}^{2}c_{B}^{2}\left[1+N\left(\mu \right)\cos2\left(\phi_{A}+\phi_{B}\right)\right]+\frac{1}{2}s_{A}^{2}s_{B}^{2}\left[1-N\left(\mu \right)\right]\right\}\\ +\eta_{2}^{k}\left\{\frac{1}{2}c_{A}^{2}s_{B}^{2}\left[1+N\left(\mu \right)\cos\left(2\phi_{A}\right)\right]+\frac{1}{2}s_{A}^{2}c_{B}^{2}\left[1-N\left(\mu \right)\cos\left(2\phi_{B}\right)\right]\right\}\\ -\eta_{2}^{k}\left\{s_{A}c_{A}s_{B}c_{B}M\left(\mu \right)\cos\left(\phi_{A}\right)\sin\left(\phi_{B}\right)\right\}\\ +\eta_{3}^{k}\left\{\frac{1}{2}c_{A}^{2}s_{B}^{2}\left[1-N\left(\mu \right)\cos\left(2\phi_{A}\right)\right]+\frac{1}{2}s_{A}^{2}c_{B}^{2}\left[1+N\left(\mu \right)\cos\left(2\phi_{B}\right)\right]\right\}\\ -\eta_{3}^{k}\left\{s_{A}c_{A}s_{B}c_{B}M\left(\mu \right)\sin\left(\phi_{A}\right)\cos\left(\phi_{B}\right)\right\}+\eta_{4}^{k}\left\{\frac{1}{2}c_{A}^{2}c_{B}^{2}\left[1-N\left(\mu \right)\cos2\left(\phi_{A}+\phi_{B}\right)\right]\right\}\\ +\eta_{4}^{k}\left\{\frac{1}{2}s_{A}^{2}s_{B}^{2}\left[1+N\left(\mu \right)\right]+s_{A}c_{A}s_{B}c_{B}M\left(\mu \right)\sin\left(\phi_{A}+\phi_{B}\right)\right\} \end{array}$$
for k=A,B and μ∈[0,1]. Where M(μ) and N(μ) are functions quantifying the noise in the quantum game, given by
(6)$$M\left(\mu \right)\equiv 2{\left(1-\mu \right)}^{2}\;;\;N\left(\mu \right)\equiv {\left(1-\mu \right)}^{4}$$
We have thus obtained the Eisert quantum game with noise played with quantum strategies when using the strategy operator (Equation 4). If we choose μ=0 in (Equation 5), we obtain the noiseless Eisert quantum game for a two-parameter set of strategies [2,3].

When evaluating $\phi_{A}=\phi_{B}=\beta_{A}=\beta_{B}=0$ in Equation (Equation 64) (see Appendix B), we obtain the classical game, i.e., the Eisert quantum game for a one-parameter set of strategies, whose strategy operator is (Equation 4). The expression corresponding to pay-offs is
(7)$$\begin{array}{c} \Pi_{k}\left(\theta_{A},\theta_{B},\mu \right)=\eta_{1}^{k}\left\{\frac{1}{2}c_{A}^{2}c_{B}^{2}\left[1+W\left(\mu \right)\right]+\frac{1}{2}s_{A}^{2}s_{B}^{2}\left[1-W\left(\mu \right)\right]\right\}\\ \;\;\;\;+\eta_{2}^{k}\left\{\frac{1}{2}c_{A}^{2}s_{B}^{2}\left[1+W\left(\mu \right)\right]+\frac{1}{2}s_{A}^{2}c_{B}^{2}\left[1-W\left(\mu \right)\right]\right\}+\eta_{3}^{k}\left\{\frac{1}{2}c_{A}^{2}s_{B}^{2}\left[1-W\left(\mu \right)\right]+\frac{1}{2}s_{A}^{2}c_{B}^{2}\left[1+W\left(\mu \right)\right]\right\}\\ \;\;\;\;\;\;\;\;\;\;\;\;\;\;\;\;\;\;\;\;+\eta_{4}^{k}\left\{\frac{1}{2}c_{A}^{2}c_{B}^{2}\left[1-W\left(\mu \right)\right]+\frac{1}{2}s_{A}^{2}s_{B}^{2}\left[1+W\left(\mu \right)\right]\right\}. \end{array}$$
We have thus obtained in (Equation 7) the Eisert quantum game with noise played with classical strategies when using the strategy operator (Equation 4) for k=A,B and μ∈[0,1]. Where W(μ) is a function corresponding to noise in a classical game, which is defined as
(8)$$W\left(\mu \right)\equiv {\left(1-\mu \right)}^{4}.$$
Equations (Equation 5) and (Equation 7) correspond to two versions of Eisert’s quantum games with noise, according to the set of strategies that players use either: *classical strategies,* i.e., a one-parameter set of strategies or *quantum strategies,* i.e., a two-parameter set of strategies).

## 3. Non-Factorizable Probabilities for Quantum Games with Noise

Quantum games and joint probabilities establish a connection with the game scheme proposed by [18]. In the previous work, they use pay-off relations that describe the factorizability of joint probabilities through a set of equations that connect player strategies with a probability distribution of player pay-offs. In this way, a quantum game can then be described as a game in which non-factorizable probabilities are the norm [6]. The pay-offs are obtained from the joint probabilities. In order to obtain this description of the quantum game, the pay-offs that describe the factorizability of the joint probabilities are used through a set of equations that connect the player ’strategies with a probability distribution of the players’ pay-offs. In this way, from the joint probabilities, we have a set of joint probabilities that may not be factorizable. Thus, a quantum game can be described as a game in which non-factorizable probabilities are allowed.

In this section, we propose a scheme based on non-factorizable probabilities for quantum games with noise, inspired by the work of reference [18]. We introduce the noise parameter μ so that when this parameter is null, we recover the expressions derived in [18].

A bimatrix description of the game is considered [6],
(9)
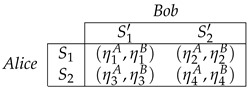

where Alice and Bob are the players, with their strategies $S_{1},S_{2}$ and $S_{1}^{\prime },S_{2}^{\prime }$, respectively, and $\eta_{1}^{k},\eta_{2}^{k},\eta_{3}^{k},\eta_{4}^{k}$ where k∈A,B, corresponding to pay-offs for each strategy of the players. For this bimatrix (Equation 9), if Alice and Bob take one strategy, e.g., $\left(S_{2},S_{1}^{\prime }\right)$, their pay-offs are $\left(\eta_{3}^{A},\eta_{3}^{B}\right)$, similarly, it happens for the other strategies.

The pay-offs for a QG with two-player noise are written as
(10)$$\Pi_{k}\left(P,Q,\mu \right)=\eta_{1}^{k}\epsilon_{1}^{R}\left(P,Q,\mu \right)+\eta_{2}^{k}\epsilon_{2}^{R}\left(P,Q,\mu \right)+\eta_{3}^{k}\epsilon_{3}^{R}\left(P,Q,\mu \right)+\eta_{4}^{k}\epsilon_{4}^{R}\left(P,Q,\mu \right)$$
where k∈A,B and μ∈[0,1]. In addition, we define $\epsilon_{i}^{R}\left(P,Q,\mu \right)$ for 1≤i≤4 as
(11)$$\left.\begin{array}{c} \epsilon_{1}^{R}\left(P,Q,\mu \right)\equiv |\langle S_{1}S_{1}^{\prime }|\psi_{f}\rangle {|}^{2}\;;\;\epsilon_{2}^{R}\left(P,Q,\mu \right)\equiv {\left|\langle S_{1}S_{2}^{\prime }|\psi_{f}\rangle \right|}^{2}\\ \epsilon_{3}^{R}\left(P,Q,\mu \right)\equiv |\langle S_{2}S_{1}^{\prime }|\psi_{f}\rangle {|}^{2}\;;\;\epsilon_{4}^{R}\left(P,Q,\mu \right)\equiv {\left|\langle S_{2}S_{2}^{\prime }|\psi_{f}\rangle \right|}^{2} \end{array}\right\}$$
These four quantum probabilities are obtained using the quantum probability rule. The probabilities $\epsilon_{i}\left(P,Q,\mu \right)$ are normalized
(12)$$\sum_{i=1}^{4}\epsilon_{i}^{R}\left(P,Q,\mu \right)=1.$$
The strategies of the players *P* and *Q* in the terms of the probabilities $\epsilon_{i}^{R}\left(P,Q,\mu \right)$ can be expressed as
(13)$$\begin{array}{ccc} P^{R} & \equiv & \epsilon_{1}^{R}\left(P,Q,\mu \right)+\epsilon_{2}^{R}\left(P,Q,\mu \right),\\ Q^{R} & \equiv & \epsilon_{1}^{R}\left(P,Q,\mu \right)+\epsilon_{3}^{R}\left(P,Q,\mu \right). \end{array}$$
Using the expressions (Equation 11), we can write the previous expressions as
(14)$$\begin{array}{ccc} P^{R} & = & |\langle S_{1}S_{1}^{\prime }|\psi_{f}\rangle {|}^{2}+{\left|\langle S_{1}S_{2}^{\prime }|\psi_{f}\rangle \right|}^{2}\\ Q^{R} & = & |\langle S_{1}S_{1}^{\prime }|\psi_{f}\rangle {|}^{2}+{\left|\langle S_{2}S_{1}^{\prime }|\psi_{f}\rangle \right|}^{2} \end{array}$$
where $P^{R}$ and $Q^{R}$ we will call *the probabilities of the game*. On the other hand, each player has the freedom to play the *P* and *Q* strategies, as these are considered independent of each other. Thus, for the pay-offs (Equation 10), the strategies of the players are P,Q∈[0,1] and not the unitary transformations $U_{A}$ and $U_{B}$. In addition, the pay-offs of the two players are linked to each other through the function $\epsilon_{1}^{R}=\epsilon_{1}^{R}\left(P,Q,\mu \right)$. As seen in [18], a classical game is one where $\epsilon_{1}=PQ$. However, for a game with noise, this restriction is not satisfied, and we say that the game is a non-factorizable game or a game that allows for non-factorizable probabilities. This type of game is called a quantum game since non-factorizability is an essential feature of an information system with an EPR configuration [16,17].

From this perspective, an immediate question for the pay-offs (Equation 10), is: what are the restrictions on the functions $\epsilon_{1}^{R}\left(P,Q,\mu \right)$? To determine this, one notices that the allowable ranges of player pay-offs in [18], necessarily must be met, i.e., $\sum_{i=1}^{4}\epsilon_{i}\left(P,Q\right)=1$, for this case is satisfied, since P,Q,μ∈[0,1]. We still have to know what the restrictions are for $\epsilon_{1}^{R}\left(P,Q,\mu \right)$. To this end, it is required that
(15)$$\epsilon_{1}^{R}\left(P,Q,\mu \right)\leq P^{R}\;;\;\epsilon_{1}^{R}\left(P,Q,\mu \right)\leq Q^{R}\;;\;\epsilon_{1}^{R}\left(P,Q,\mu \right)\leq P^{R}+Q^{R}.$$
Then, the pay-offs (Equation 10) are equivalent to those of the quantum game in reference [18], for μ=0. Referring to the expressions (Equation 10), we obtain that for given *P*, Q∈[0,1] as independent player strategies, the function $\epsilon_{1}^{R}\left(P,Q,\mu \right)$ gives a value in [0,1], the functions $\epsilon_{2}^{R}\left(P,Q,\mu \right)$, $\epsilon_{3}^{R}\left(P,Q,\mu \right)$ and $\epsilon_{4}^{R}\left(P,Q,\mu \right)$ which are defined as
(16)$$\begin{array}{ccc} \epsilon_{2}^{R}\left(P,Q,\mu \right) & \equiv & P^{R}-\epsilon_{1}^{R}\left(P,Q,\mu \right);\\ \epsilon_{3}^{R}\left(P,Q,\mu \right) & \equiv & Q^{R}-\epsilon_{1}^{R}\left(P,Q,\mu \right) \end{array}$$
and
(17)$$\epsilon_{4}^{R}\left(P,Q,\mu \right)\equiv 1-\left[\left(P^{R}+Q^{R}\right)-\epsilon_{1}^{R}\left(P,Q,\mu \right)\right].$$
When restrictions (Equation 15) hold, the functions $\epsilon_{2}^{R}\left(P,Q,\mu \right)$, $\epsilon_{3}^{R}\left(P,Q,\mu \right)$ and $\epsilon_{4}^{R}\left(P,Q,\mu \right)$ produce values within the range [0,1] and when μ=0, the non-factorizable probabilities for quantum games are retrieved, In view of the above, for a non-factorizable game with noise, it is required that $\epsilon_{1}^{R}\left(P,Q,\mu \right)$, is restricted by (Equation 15) and by the condition
(18)$$\epsilon_{1}^{R}\left(P,Q,\mu \right)\leq PQ.$$
Therefore, the condition that guarantees the non-factorizability of a game with noise is limited by that of a game without noise since the restrictions are the same [6,18].

## 4. Joint Probabilities for Quantum Games with Noise

In the previous section, we developed the joint probabilities approach to Eisert QG [2,3], now including noise for the first time. For this purpose, we wrote the parameters $\epsilon_{i}^{R}$, the game probabilities (Equation 23), and we determined the NE inequalities for Eisert QG with noise, where the pay-offs are associated with two versions of the game (Equation 5) and (Equation 7), which we wrote according to the set of strategies that players use as *Noise quantum game with classic strategies* (correspond to a one-parameter set of strategies) and *Noise quantum game with quantum strategies* (correspond to a two-parameter set of strategies), where the super-index 🟉 represents the quantum strategies. Also, we determine the restrictions to which the Nash equilibrium is subject.

Based on the expression obtained in (Equation 7), for noise quantum games with classic strategies, we have the parameters $\epsilon_{i}^{R}$ written as
(19)$$\epsilon_{1}^{R}=\frac{1}{2}\left(c_{A}^{2}c_{B}^{2}+s_{A}^{2}s_{B}^{2}\right)+\frac{1}{2}\left(c_{A}^{2}c_{B}^{2}-s_{A}^{2}s_{B}^{2}\right)W\left(\mu \right);$$
(20)$$\epsilon_{2}^{R}=\frac{1}{2}\left(c_{A}^{2}s_{B}^{2}+s_{A}^{2}c_{B}^{2}\right)+\frac{1}{2}\left(c_{A}^{2}s_{B}^{2}-s_{A}^{2}c_{B}^{2}\right)W\left(\mu \right),$$
(21)$$\epsilon_{3}^{R}=\frac{1}{2}\left(c_{A}^{2}s_{B}^{2}+s_{A}^{2}c_{B}^{2}\right)-\frac{1}{2}\left(c_{A}^{2}s_{B}^{2}-s_{A}^{2}c_{B}^{2}\right)W\left(\mu \right);$$
(22)$$\epsilon_{4}^{R}=\frac{1}{2}\left(c_{A}^{2}c_{B}^{2}+s_{A}^{2}s_{B}^{2}\right)-\frac{1}{2}\left(c_{A}^{2}c_{B}^{2}-s_{A}^{2}s_{B}^{2}\right)W\left(\mu \right),$$
where the game probabilities are
(23)$$P^{R}=\epsilon_{1}^{R}+\epsilon_{2}^{R}\;;\;Q^{R}=\epsilon_{1}^{R}+\epsilon_{3}^{R},$$
defining the probabilities [6,18],
(24)$$p\equiv \cos^{2}\left(\frac{\theta_{A}}{2}\right)=c_{A}^{2}\;;\;q\equiv \cos^{2}\left(\frac{\theta_{B}}{2}\right)=c_{B}^{2},$$
where p,q∈[0,1]. Therefore, using these definitions in (Equation 19) and (Equation 21), we have
(25)$$\epsilon_{1}^{R}=\frac{1}{2}\left(2pq-p-q+1\right)+\frac{1}{2}\left(p+q-1\right)W\left(\mu \right);$$
(26)$$\epsilon_{2}^{R}=\frac{1}{2}\left(p+q-2pq\right)+\frac{1}{2}\left(p-q\right)W\left(\mu \right),$$
(27)$$\epsilon_{3}^{R}=\frac{1}{2}\left(p+q-2pq\right)-\frac{1}{2}\left(p-q\right)W\left(\mu \right);$$
(28)$$\epsilon_{4}^{R}=\frac{1}{2}\left(2pq-p-q+1\right)-\frac{1}{2}\left(p+q-1\right)W\left(\mu \right).$$
The game probabilities are obtained by entering the appropriate expressions in (Equation 23)
(29)$$P^{R}=\frac{1}{2}\left[1+(2p-1)W(\mu )\right]\;,\;Q^{R}=\frac{1}{2}\left[1+(2q-1)W(\mu )\right].$$
In the expression (Equation 25), we see that the joint probabilities in the parameter $\epsilon_{1}^{R}$ are a linear combination of the probabilities for a classical game, plus the presence of W(μ) which is a function of noise. Therefore, the QG with classical strategies, when incorporating the noise, ceases to behave as a classical game from the point of view of pay-offs since correlations can be enhanced, which makes the joint probabilities in $\epsilon_{1}^{R}$ non-factorizable.

By means of the approach of the previous section, we obtain a specific description of the noisy QG with quantum strategies. Taking the expression (Equation 5), we identify the parameters $\epsilon_{i}^{★R}$ and using the trigonometric identities $\cos\left(2x\right)=2\cos^{2}\left(x\right)-1$ y sin(xy)=sin(x)cos(y)sin(y)cos(x), we have
(30)$$\begin{array}{cc} \epsilon_{1}^{★R} & =\epsilon_{1}^{★}N\left(\mu \right)+\frac{1}{2}\left(c_{A}^{2}c_{B}^{2}+s_{A}^{2}s_{B}^{2}\right)\left(1-N\left(\mu \right)\right), \end{array}$$
(31)$$\begin{array}{cc} \epsilon_{2}^{★R} & =\frac{1}{2}c_{A}^{2}c_{B}^{2}\left[1+N\left(\mu \right)\left(2\cos^{2}\phi_{A}-1\right)\right]+\frac{1}{2}s_{A}^{2}c_{B}^{2}\left[1-N\left(\mu \right)\left(2\cos^{2}\phi_{B}-1\right)\right]\\ \; & \;-s_{A}c_{A}s_{B}c_{B}M\left(\mu \right)\cos\phi_{A}\sin\phi_{B}, \end{array}$$
(32)$$\begin{array}{cc} \epsilon_{3}^{★R} & =\frac{1}{2}c_{A}^{2}c_{B}^{2}\left[1-N\left(\mu \right)\left(2\cos^{2}\phi_{A}-1\right)\right]+\frac{1}{2}s_{A}^{2}c_{B}^{2}\left[1+N\left(\mu \right)\left(2\cos^{2}\phi_{B}-1\right)\right]\\ \; & \;-s_{A}c_{A}s_{B}c_{B}M\left(\mu \right)\sin\phi_{A}\cos\phi_{B}, \end{array}$$
(33)$$\begin{array}{cc} \epsilon_{4}^{★R} & =\frac{1}{2}\left(c_{A}^{2}c_{B}^{2}+s_{A}^{2}s_{B}^{2}\right)\left(1+N\left(\mu \right)\right)-\epsilon_{1}^{★}N\left(\mu \right)\\ \; & \;+s_{A}c_{A}s_{B}c_{B}M\left(\mu \right)\left(\sin\phi_{A}\cos\phi_{B}+\cos\phi_{A}\sin\phi_{B}\right). \end{array}$$
where $\epsilon_{1}^{★}={\left(1-2u\right)}^{2}w^{2}$ (see [6]), the game probabilities are given by the expressions
(34)$$P^{★R}=\epsilon_{1}^{★R}+\epsilon_{2}^{★R}\;;\;Q^{★R}=\epsilon_{1}^{★R}+\epsilon_{3}^{★R}$$
defining the probabilities [6],
(35)$$p^{★}\equiv \cos^{2}\phi_{A}\;;\;q^{★}\equiv \cos^{2}\phi_{B},$$
where evidently to $p^{★},q^{★}\in \left[0,1\right]$. Using the definitions (Equation 24) and (Equation 35) in expressions (Equation 30) to (33), we have the parameters $\epsilon_{i}^{★R}$ written in terms of probabilities, as follows
(36)$$\epsilon_{1}^{★R}=\frac{1}{2}\left(2pq-p-q+1\right)-\frac{1}{2}\left(2pq-p-q+1-2\epsilon_{1}^{★}\right)N\left(\mu \right),$$
(37)$$\epsilon_{2}^{★R}=\frac{1}{2}\left(p+q-2pq\right)+\left(p^{★}-q^{★}\right)N\left(\mu \right)-\sqrt{pq\left(1-p\right)\left(1-q\right)p^{★}\left(1-q^{★}\right)}M\left(\mu \right),$$
(38)$$\epsilon_{3}^{★R}=\frac{1}{2}\left(p+q-2pq\right)-\left(p^{★}-q^{★}\right)N\left(\mu \right)-\sqrt{pq\left(1-p\right)\left(1-q\right)p^{★}\left(1-q^{★}\right)}M\left(\mu \right),$$
(39)$$\begin{array}{c} \epsilon_{4}^{★R}=\frac{1}{2}\left(2pq-p-q+1\right)+\frac{1}{2}\left(2pq-p-q+1-2\epsilon_{1}^{★}\right)N\left(\mu \right)\\ \;\;\;\;\;\;\;\;+\sqrt{pq(1-p)(1-q)}\left[\sqrt{p^{★}\left(1-q^{★}\right)}+\sqrt{\left(1-p^{★}\right)q^{★}}\right]M\left(\mu \right), \end{array}$$
and the game probabilities are written as
(40)$$\begin{array}{cc} P^{★R}=\frac{1}{2}+\frac{1}{2}[2\left(p^{★}-q^{★}\right) & \;-2pq+p+q-1+2\epsilon_{1}^{★}]N\left(\mu \right)\\ \; & \;-\sqrt{pq\left(1-p\right)\left(1-q\right)p^{★}\left(1-q^{★}\right)}M\left(\mu \right), \end{array}$$
(41)$$\begin{array}{cc} Q^{★R}=\frac{1}{2}+\frac{1}{2}[2\left(q^{★}-p^{★}\right) & \;-2pq+p+q-1+2\epsilon_{1}^{★}]N\left(\mu \right)\\ \; & \;-\sqrt{pq\left(1-p\right)\left(1-q\right)\left(1-p^{★}\right)q^{★}}M\left(\mu \right). \end{array}$$

Evaluating the probability values at the Pareto and Nash equilibria, i.e., when the proposed probabilities *p*, *q*, $p^{★}$ and $q^{★}$ are equal [6], we have
(42)$$p=q\equiv w\;;\;p^{★}=q^{★}\equiv u.$$
so that the $\epsilon_{i}^{★R}$
(43)$$\epsilon_{1}^{★R}=\frac{1}{2}\left[1-2w(1-w)\right]-\frac{1}{2}\left[1-2w\left(1-w\right)-2\epsilon_{1}^{★}\right]N\left(\mu \right),$$
(44)$$\epsilon_{2}^{★R}=\epsilon_{3}^{★R}=w\left(1-w\right)\left[1-\sqrt{u(1-u)}M\left(\mu \right)\right],$$
(45)$$\begin{array}{cc} \epsilon_{4}^{★R}=\frac{1}{2}\left[1-2w(1-w)\right]\; & +\frac{1}{2}\left[1-2w\left(1-w\right)-2\epsilon_{1}^{★}\right]N\left(\mu \right)\\ \; & +2w\left(1-w\right)\sqrt{u(1-u)}M\left(\mu \right). \end{array}$$
and the game probabilities are
(46)$$P^{★R}=\frac{1}{2}+\frac{1}{2}\left[2w\left(1-w\right)+2\epsilon_{1}^{★}-1\right]-w\left(1-w\right)\sqrt{u(1-u)}M\left(\mu \right),$$
(47)$$Q^{★R}=\frac{1}{2}+\frac{1}{2}\left[2w\left(1-w\right)+2\epsilon_{1}^{★}-1\right]-w\left(1-w\right)\sqrt{u(1-u)}M\left(\mu \right).$$
From the parameter, $\epsilon_{1}^{★R}$ pointed out at (Equation 43), we conclude that the noise does not destroy the non-factorizable nature of the joint probabilities for QG (the term $\epsilon_{1}^{★}$ remains), that is, noise does not destroy entanglement. On the other hand, It’s easy to see when there is no presence of noise at (Equation 43), that is μ=0, we obtain $\epsilon_{1}^{★}$, which is consistent with the result obtained in [6]. Furthermore, a new relationship associated with noise appears through the presence of the function N(μ). Thus, a noise quantum game with quantum strategies is still a QG since quantum correlations are not lost in the joint probabilities.

### 4.1. Nash Equilibrium in Joint Probabilities for Quantum Games with Noise

In order to obtain a complete description of QG with noise through their joint probabilities, it is necessary to determine the NE inequalities with noise corresponding to our approach. For this purpose, we follow the formulation of inequalities proposed in [18] for NE. In general, the NE for a QG with noise must satisfy the following inequalities
(48)$$\Pi_{A}\left(P^{NE},Q^{NE},\mu \right)-\Pi_{A}\left(P,Q^{NE},\mu \right)\geq 0\;;\;\Pi_{B}\left(P^{NE},Q^{NE},\mu \right)-\Pi_{B}\left(P^{NE},Q,\mu \right)\geq 0,$$
for the expressions (Equation 10), where the strategy $\left(P^{NE},Q^{NE}\right)$ defines NE. On the other hand, when μ=0, the NE inequalities for the scheme without noise [6,18] are obtained. In this way, we determine that the NE inequalities must satisfy (Equation 48), for the proposed scheme of joint probabilities with classical and quantum strategies.

Now, we write the pay-offs for Alice and Bob with classical strategies. Starting from the parameters $\epsilon_{i}$ of the Equations (Equation 25) and (Equation 27), we have
(49)$$\begin{array}{cc} \Pi_{k}\left(p,q,\mu \right) & =\frac{\eta_{1}^{k}}{2}\left(2pq-p-q+1\right)+\frac{\eta_{1}^{k}}{2}\left(p+q-1\right)W\left(\mu \right)+\frac{\eta_{2}^{k}}{2}\left(p+q-2pq\right)\\ \; & +\frac{\eta_{2}^{k}}{2}\left(p-q\right)W\left(\mu \right)+\frac{\eta_{3}^{k}}{2}\left(p+q-2pq\right)-\frac{\eta_{3}^{k}}{2}\left(p-q\right)W\left(\mu \right)\\ \; & \;+\frac{\eta_{4}^{2}}{2}\left(2pq-p-q+1\right)-\frac{\eta_{4}^{k}}{2}\left(p+q-1\right)W\left(\mu \right), \end{array}$$
From the expressions indicated in (Equation 48) and using the pay-off (Equation 49), we obtain the following expressions
(50)$$\left[\left(\eta_{1}^{A}-\eta_{2}^{A}-\eta_{3}^{A}+\eta_{4}^{A}\right)\left(2q^{NE}-1\right)+\left(\eta_{1}^{A}+\eta_{2}^{A}-\eta_{3}^{A}-\eta_{4}^{A}\right)W\left(\mu \right)\right]\left(p^{NE}-p\right)\geq 0,$$
(51)$$\left[\left(\eta_{1}^{B}-\eta_{2}^{B}-\eta_{3}^{B}+\eta_{4}^{B}\right)\left(2p^{NE}-1\right)+\left(\eta_{1}^{B}-\eta_{2}^{B}+\eta_{3}^{B}-\eta_{4}^{B}\right)W\left(\mu \right)\right]\left(q^{NE}-q\right)\geq 0.$$
These inequalities are what the Nash equilibrium must satisfy when the chosen strategy corresponds is $\left(p^{NE},q^{NE}\right)$.

Analogous to the classical case, we use Equations (Equation 43)–(Equation 45), to compute the pay-offs for both Alice and Bob as
(52)$$\begin{array}{cc} \Pi_{k}^{★}\left(w,u,\mu \right) & =\frac{\eta_{1}^{k}}{2}\left[1-2w(1-w)\right]-\frac{\eta_{1}^{k}}{2}\left[1-2w\left(1-w\right)-2\epsilon_{1}^{★}\right]N\left(\mu \right)\\ \; & +\left(\eta_{2}^{k}+\eta_{3}^{k}\right)w\left(1-w\right)\left[1-\sqrt{u(1-u)}M\left(\mu \right)\right]+\frac{\eta_{4}^{k}}{2}\left[1-2w(1-w)\right]\\ \; & +\frac{\eta_{4}^{k}}{2}\left[1-2w\left(1-w\right)-2\epsilon_{1}^{★}\right]N\left(\mu \right)+2\eta_{4}^{k}w\left(1-w\right)\sqrt{u(1-u)}M\left(\mu \right). \end{array}$$
Using the NE conditions (Equation 48) and the pay-offs (Equation 52), we obtain the following equations
(53)$$\begin{array}{cc} \; & \;\left\{\left(\eta_{1}^{A}+\eta_{4}^{A}\right)\left(w^{NE}+w-1\right)+\left(\eta_{1}^{A}-\eta_{4}^{A}\right)\left[\left(1-w^{NE}-w\right)+{\left(1-2u^{NE}\right)}^{2}\left(w^{NE}+w\right)\right]N\left(\mu \right)\right.\\ \; & \;+\left.\left(1-w^{NE}-w\right)\left(\eta_{2}^{A}+\eta_{3}^{A}\right)\right\}\left(w^{NE}-w\right)\\ \;- & \left(1-w^{NE}-w\right)\left(\eta_{2}^{A}+\eta_{3}^{A}-2\eta_{4}^{A}\right)\sqrt{u^{NE}\left(1-u^{NE}\right)}M\left(\mu \right)\left(w^{NE}-w\right)\geq 0, \end{array}$$
(54)$$\begin{array}{cc} \; & \;4\left(\eta_{1}^{A}-\eta_{4}^{A}\right)\left(u^{NE}-u\right)\left(u^{NE}+u-1\right){\left(w^{NE}\right)}^{2}N\left(\mu \right)\\ \; & \;+\left(\eta_{2}^{B}+\eta_{3}^{B}-2\eta_{4}^{B}\right)w^{NE}\left(1-w^{NE}\right)\left[\sqrt{u(1-u)}-\sqrt{u^{NE}\left(1-u^{NE}\right)}\right]M\left(\mu \right)\geq 0. \end{array}$$
The above inequalities must be satisfied for $\left(w^{NE},u^{NE}\right)$ for the quantum strategies.

### 4.2. Quantum Games with Noise and Their Nash Equilibria

At this point, we determine the restrictions imposed by the NE on the games: the Prisoner’s Dilemma, the Chicken Game, and the Battle of the Sexes, using the expressions (Equation 50), (Equation 51) and (Equation 53), (54), which correspond to the classical and quantum strategies, respectively, for the probability values corresponding to the strategies that generate the said equilibrium in the absence of noise Table 1. The restrictions imposed by the NE on three games are shown in Table 2.

In Table 1, the probability values for which NE is achieved in the three games are shown. As we see, the NE for the Prisoner’s Dilemma and the Chicken Game are obtained for the same probability value, i.e., when quantum strategies are employed [2,3]. In contrast, for the Battle of the Sexes, there are infinite Nash equilibria when quantum strategies are considered [27]. Among these equilibria is the one that corresponds to the values associated with the parameters $\theta_{A}=\theta_{B}=\pi$ and $\phi_{A}=\phi_{B}=\pi /4$, equivalent to the probability values $\left(w_{3}^{N},u_{3}^{N}\right)=\left(0,1/2\right)$.

Where the probability values for NE’s are $\left(p_{1}^{N},q_{1}^{N}\right)=\left(0,0\right)$, $\left(p_{2}^{N},q_{2}^{N}\right)=\left(1,0\right)$, $\left(p_{3}^{N},q_{3}^{N}\right)=\left(0,1\right)$, $\left(p_{4}^{N},q_{4}^{N}\right)=\left(1,1\right)$ and $\left(w_{1}^{N},u_{1}^{N}\right)=\left(1,0\right)$, $\left(w_{2}^{N},u_{2}^{N}\right)=\left(1,1\right)$, $\left(w_{3}^{N},u_{3}^{N}\right)=\left(0,1/2\right)$, for classical and quantum strategies, respectively [6].

From Table 2, conditions were obtained on the parameter μ that limits the NE. For the Prisoner’s Dilemma the values of $\mu =1-\sqrt[{4}]{\frac{1}{3}}∼0,2$ and $\mu =1-\frac{1}{\sqrt[{4}]{2}}∼0,2$. We note, for the approach we take in this work, our results differ from those obtained by [23], for the NE of the Prisoner’s Dilemma, because we take a unique parameter for noise ($\mu_{A}=\mu_{B}=\mu$), which is different to the approach taken in Ref. [23], that takes different parameters for noise to the players ($\mu_{A}\neq \mu_{B}$).

For the Chicken Game, the NE inequalities do hold $\mu =\sqrt[{4}]{\frac{2}{3}}∼0,1$ as the threshold value of noise for quantum strategies, again in contrast to the case of the Battle of the Sexes where there are no restrictions for both strategies.

## 5. The Symmetry of Joint Probabilities in Quantum Games with Noise

In this section, we obtain the parameters $\epsilon_{i}^{R}$ and the probabilities for the games considered. From the terms $\epsilon_{1}^{R}$ and $\epsilon_{1}^{★R}$, the non-factorizability of the joint probabilities for these games remains. To show this, we use the equations corresponding to the joint probabilities of the previous section for each of the games the Prisoner’s Dilemma, the Chicken Game, and the Battle of the Sexes, as well as the corresponding pay-off matrix for each one (see Appendix A). Depending on the pay-off values corresponding to each game, we indicate the presence of the parameter $\epsilon_{i}^{R}$ by one and its absence by zero in the equations for pay-offs. In addition, the super-index 🟉 represents the use of quantum strategies. The results for the games are shown in Table 3 and Table 4.

For the Prisoner’s Dilemma, we see the anti-symmetry, under the exchange of players, in the game probabilities in Table 4. The game probabilities for the value of $P^{R(DP)}$ for Alice is equal to the value of $Q^{R(DP)}$ for Bob. The same occurs for the value of $Q^{R(DP)}$ for Alice with the value of $P^{\left(DP\right)}$ for Bob. Likewise, for the value of $\epsilon_{2}^{R(DP)}=0$ for Alice and $\epsilon_{3}^{R(DP)}=0$ for Bob (see Table 3). The same occurs for the quantum case. With the Chicken Game, you have the symmetry of the game odds for Alice and Bob in Table 4. Also, the terms $\epsilon_{4}^{R(CG)}$ and $\epsilon_{4}^{★R(CG)}$ are null for both players (see Table 3). The aforementioned symmetry is due to the fact that the rest of the game’s parameters are the same for both players.

An important fact for classical strategies is that each player’s game probabilities will differ since they are obtained from the expressions (Equation 29). This is the opposite of when quantum strategies are used since the probabilities for each player are equal and correspond to the expressions obtained in (Equation 46) and (Equation 47). This is because the correlations of the joint probabilities are due to entanglement. The symmetry discussed above will be called a weak symmetry of the game probabilities because they are the same for quantum strategies and different for classical strategies. Looking at the Battle of the Sexes in Table 4, we see that the probabilities, besides being symmetrical, are the same game probabilities for the players, that is, $P^{R(BS)}=Q^{R(BS)}$ and $P^{★R(BS)}=Q^{★R(BS)}$, in particular, this strong symmetry is because there is an absence of the parameters $\epsilon_{2}^{R(BS)}$, $\epsilon_{3}^{R(BS)}$ and $\epsilon_{2}^{★R(BS)}$, $\epsilon_{3}^{★R(BS)}$ since these are null as we see in the Table 3. In the same way, when focusing on classical strategies, since the same reasoning is valid for quantum strategies, the game probabilities are determined only by the term $\epsilon_{1}^{R(BS)}$, that is to say, $P^{R(BS)}=Q^{R(BS)}=\epsilon_{1}^{R(BS)}$. We consider this symmetry a strong symmetry of the game probabilities since the game probabilities are the same for both strategies. For the three games, in addition to having analyzed the parameters $\epsilon_{i}^{R}$ and how these affect the symmetry and antisymmetry of the game probabilities, we obtained that the joint probabilities for the games are non-factorizable, since the terms $\epsilon_{1}^{R}$ and $\epsilon_{1}^{★R}$, are present as we see in Table 3.

## 6. Correlation Measure from Joint Probabilities and Entanglement in Quantum Games

The measure of entanglement in different physical contexts is still a current problem [28,29,30,31,32,33]. Different types of entanglement measures have been proposed [19,20,21], for different approaches to quantifying entanglement [21], such as (i) computing a correlation that contradicts the theory of elements of reality, (ii) evaluating the global structure of the wave function, (iii) sensitivity to interference with secure communication, (iv) application of positive maps in physics, and (v) quantifying a correlation that is stronger than any classical correlations. Nevertheless, an entanglement measure has not been devised, to our knowledge, for quantum games. Throughout this work, we have discussed how we understand entanglement as strong correlations of the joint probabilities. From this fact, we can quantify the entanglement in our game approach through a correlation measure.

Here, we propose a correlation measure based on the joint probabilities and the information of the games. The measure proposed is determined from the information quantified by the term $\epsilon_{1}^{\alpha }$ since this term characterizes classical and quantum games depending on whether their joint probabilities are factorizable or not. Therefore, we name it: entanglement parameter probability.

The correlation measure that we propose is defined as
(55)$$\gamma \equiv \Lambda \;J\left(\epsilon_{1}^{\alpha }\right)=\lim_{w,u\to w^{\gamma },u^{\gamma }}\;H\left(\epsilon_{1}^{\alpha }\right),$$
where α represents the strategies to consider, either classical or quantum. The quantum strategies are represented by the super-index 🟉 and the classical without the super-index. In this way, we name $\epsilon_{1}^{★}$ the entanglement probability. Also, $w^{\gamma }$ and $u^{\gamma }$ are defined as the values where the entanglement produces an upper bound of information, that is, the values for which the information is a maximum for the term $\epsilon_{4}^{★}$ (see [6]). Since the latter term shows the correlation of the input and output information of the game, then, being $H(\epsilon_{4}^{★})$ a maximum, it reflects the complete influence of entanglement in the game when one uses quantum strategies. The values for $w^{\gamma }$ and $u^{\gamma }$ are obtained from
(56)$$\Phi \equiv \max_{w^{\gamma },u^{\gamma }}\;J\left(\epsilon_{4}^{★}\right)=\max_{w^{\gamma },u^{\gamma }}\;H\left[\left[1+4u\left(1-u\right)\right]w^{2}+1-2w+4w\left(1-w\right)\sqrt{u(1-u)}\right]=1,$$
whose values of probabilities are $w_{\left(1\right)}^{\gamma }=1$; $u_{\left(1\right)}^{\gamma }=\frac{1}{2}\left(1+\frac{1}{\sqrt{2}}\right)$ or $w_{\left(2\right)}^{\gamma }=1$; $u_{\left(2\right)}^{\gamma }=\frac{1}{2}\left(1-\frac{1}{\sqrt{2}}\right)$.

Two consequences of incorporating entropy as the correlation measure are (Equation 55): First, we have a measure that uses the nature of joint probabilities to determine the correlation in quantum games which closely related to entanglement. Second, we naturally obtain a normalized measure, that is, 0≤γ≤1. Therefore, we have the values that represent the minimum and maximum correlation for our definition (Equation 55).

With this in mind, we proceed to determine the correlation measure for classical and quantum strategies. For these, it does not matter which of the values use for solutions of $w^{\gamma }$ and $u^{\gamma }$ since the same results ensue.

Thus, the correlation measure for classical and quantum strategies are as follows
(57)$$\begin{array}{cc} \gamma & =\Lambda \;J\left(\epsilon_{1}\right)=\lim_{w,u\to w^{\gamma },u^{\gamma }}H\left(w^{2}\right)=0, \end{array}$$
(58)$$\begin{array}{cc} \gamma & =\Lambda \;J\left(\epsilon_{1}^{★}\right)=\lim_{w,u\to w^{\gamma },u^{\gamma }}H\left[{\left(1-2u\right)}^{2}w^{2}\right]=1. \end{array}$$
In this way, we determine that the correlation for quantum games is a maximum (Equation 58) due to the fact that the correlation of the probability is maximum, which is closely related to entanglement because it is consistent with Eisert’s quantum game approach [2,3], where maximum entanglement states are used. We also show the robustness of our definition (Equation 55), determining that the correlation in the classical games does not exist (Equation 57) but the entanglement if it exists because the initial state is maximally entangled [2,3]. Finally, we see that the strong correlations of the joint probabilities are contained in the entanglement probability parameter that reflects the entanglement quantified through the definition (Equation 55) in our game approach for players the use quantum strategies.

## 7. Correlation Measure from Joint Probabilities and Entanglement in the Quantum Games with Noise

The quantification of entanglement for different types of entanglement measures has been proposed [19,20,21]. In this work, we understand entanglement as strong correlations of the joint probabilities as in [6]. In this direction, we can quantify the entanglement for quantum games with noise through a correlation measure.

The proposal of a correlation measure based on the joint probabilities and the information of the games in [6] is determined from the information measure of the term $\epsilon_{1}^{\alpha }$ since this term characterizes and therefore classifies the games classical or quantum. If their joint probabilities are factorizable, it will be classical; otherwise, it will be quantum. For quantum games with noise, the relevant term is $\epsilon_{1}^{\alpha R}$. Then, the entanglement measure is defined as
(59)$$\gamma \equiv \Lambda \;J\left(\epsilon_{1}^{\alpha R}\right)=\lim_{w,u\to w^{\gamma },u^{\gamma }}\;H\left(\epsilon_{1}^{\alpha R}\right).$$
We have that α represents the strategies to consider, either classical or quantum. The quantum strategies are represented by the super-index 🟉 and the classical without super-index. Also, $w^{\gamma }$ and $u^{\gamma }$ are defined as the values where the entanglement produces an upper bound of information. These values of the probabilities are the same for quantum games with noise. Because we require consistency, when μ=0, therefore, values of probabilities are $w_{\left(1\right)}^{\gamma }=1$; $u_{\left(1\right)}^{\gamma }=\frac{1}{2}\left(1\frac{1}{\sqrt{2}}\right)$ or $w_{\left(2\right)}^{\gamma }=1$; $u_{\left(2\right)}^{\gamma }=\frac{1}{2}\left(1-\frac{1}{\sqrt{2}}\right)$.

As a consequence of incorporating entropy in the correlation measure (Equation 59), we naturally obtain a normalized measure, that is, 0≤γ≤1, the values represent the minimum and maximum correlation for our definition (Equation 59). In this way, we proceed to determine the correlation measure for classical and quantum strategies for quantum games with noise. For these, it does not matter which of the values one uses for the solutions of $w^{\gamma }$ and $u^{\gamma }$ since the same results are obtained. Thus, the correlation measure for classical and quantum strategies are as follows:(60)$$\begin{array}{cc} \; & \;\gamma =\Lambda \;J\left(\epsilon_{1}^{R}\right)=\lim_{w,u\to w^{\gamma },u^{\gamma }}H\left[\frac{1}{2}\left(2w^{2}-2w+1\right)+\frac{1}{2}\left(2w-1\right)W\left(\mu \right)\right]=H\left[\frac{1}{2}\left(1+W\left(\mu \right)\right)\right], \end{array}$$(61)$$\begin{array}{cc} \; & \;\gamma =\Lambda \;J\left(\epsilon_{1}^{★R}\right)=\lim_{w,u\to w^{\gamma },u^{\gamma }}H\left[\frac{1}{2}\left[1-2w(1-w)\right]-\frac{1}{2}\left[1-2w\left(1-w\right)-2\epsilon_{1}^{★}\right]N\left(\mu \right)\right]=1, \end{array}$$
where $W\left(\mu \right)={\left(1-\mu \right)}^{4}$ and $\epsilon_{1}^{★}={\left(1-2u\right)}^{2}w^{2}$. From the above expressions for the correlation measure, we determined that the correlation for quantum games with noise is maximum (Equation 61), which is consistent with the obtained (Equation 58). This result is closely related to Eisert’s quantum game approach with noise [22,23], where the states are maximally entangled. We also show the robustness of our definition (Equation 59). However, the correlations that emerge (Equation 60) in contrast with the classical games without noise (Equation 57) have interesting behavior. Apparently, incorporating the noise in quantum games with classical strategies can generate stronger correlations that are similar to quantum strategies.

Until now, we have seen that strong correlations of the joint probabilities reflect the entanglement that we have quantified through the definition (Equation 59) in our game approach. In such terms, with the purpose of dispelling any uncertainty regarding the correlation measure (Equation 60), we have determined the behavior for both expressions (Equation 60) and (Equation 61), which is depicted in Figure 1.

As the noise increases in the classical strategies, the correlations generated approximate the behavior produced by a game with quantum strategies Figure 1. In other words, the noise produces correlations that we can understand as players use quantum strategies in the classical game without the need to saturate the system with the maximum value of noise as we see in Figure 1 for our game approach.

## 8. Summary and Conclusions

We addressed the Eisert formulation of quantum games with noise. We proposed a formalism in terms of non-factorizable probabilities inspired by the proposal of reference [18]. From this formalism, quantum games with noise are connected to the description in terms of joint probabilities. In the same way, games are characterized by the nature of their joint probabilities; if these are factorizable, they are a classical game, and if they are not, they have a quantum component [17], and non-factorizability is connected to entanglement.

Our results show two interesting general facts; the first is that the presence of noise in the game does not destroy entanglement, and second, the presence of noise in a QG with classical strategies can introduce correlations in the joint probabilities that, in our approach, resemble QG with quantum strategies since the joint probabilities are not factorizable. This does not violate any principles since the QG deal in quantum amplitudes, and increased correlations come from an interplay of quantum behavior and noise.

Regarding the Nash equilibrium in the presence of noise, we find that NE can be disrupted by noise depending on the game analyzed. While for the Prisoner’s dilemma, the NE is disrupted for both quantum and classical strategies beyond a threshold value of noise, for the Chicken game no disruption occurs for classical strategies, while a threshold appears for the quantum strategies. Interestingly the Battle of the sexes suffers no disruption from the noise of either strategy. We computed exact values for the threshold when they occur in the quantum games analyzed. We tracked down the different behaviors mentioned to the symmetries in the game probabilities that can be strategy-dependent (classical or quantum). Also, we produced a classification in terms of the strength of these symmetries (strong or weak) so that they are able not to be disrupted by disorder.

On the other hand, we propose a correlation measure from previous work [6] for quantum games (Equation 55). This correlation measure is applied to quantum games with noise from their joint probabilities (Equation 59). We obtain that for quantum strategies, the maximum correlation, due to the noise, does not affect the entanglement (Equation 61). However, when classical strategies are used, we find that our model of disorder generates correlations that emulate a quantum game with maximum correlation; in other words, the incorporation of noise into quantum games with classical strategies generates correlations similar to those used by the players with quantum strategies into the classical game without the need to saturate the system with the maximum value of noise.

In addition to the realm of noise quantum games, we explored the potential of conceiving noisy quantum games as practical information channels endowed with a specific channel capacity. This notion was initially hinted at in our prior work [6], wherein we introduced the concept of an information capacity for quantum games from joint probabilities. Within this framework, quantum games from joint probabilities are cast as distinct information channels, with each individual game assuming the role of a unique channel.

In the context of the work in reference [6], we demonstrated that strategic approaches that effectively mitigate dilemmas in scenarios like the Battle of the Sexes exhibit a notably superior capacity compared to the Flip channel. Consequently, this novel approach may serve as a promising avenue toward the development of high-capacity information channels that exhibit resilience against noise. Importantly, the capacity of such a channel directly correlates with its ability to withstand the influence of noise and decoherence.

It is worth highlighting that the amalgamation of the formalism proposed for quantum games, incorporating noise from joint probabilities, with the information capacity framework delineated in [6], presents a compelling and innovative avenue for advancing the transmission of information within the realm of quantum information channels.

Other applications for games with joint probabilities are related to infinitely repeated quantum games and the efficiency of these strategies with respect to their simple counterpart, also showing the Quantum Folk Theorem [34]. Likewise, the basis for many important game theoretical models in economics are the games of extensive form in their quantum version. A recent application of these games is the quantization of the Angel problem [35]. Another interesting application is related to single qubit games with graphical analysis; one notices that the curves representing the players’ gains display a behavior similar to the curves that give rise to a phase transition in thermodynamics [36]. These transitions are associated with optimal strategy changes and occur in the absence of entanglement and interaction between the players.

Finally, quantum games with noise propose mapping game concepts to statistical mechanics describing thermalization processes using classical games [37]. Exploring the quantum game counterpart could be illuminating for identifying decoherence processes understood through quantum game equilibria as described here and in reference [6].

## Figures and Tables

**Figure 1 entropy-25-01222-f001:**
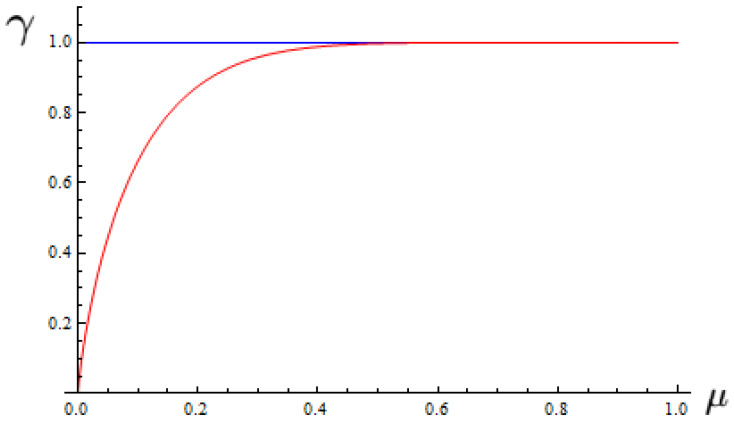
Entanglement measure for quantum games with noise from joint probabilities. Corresponding to the classical strategies (red) and quantum strategies (blue).

**Table 1 entropy-25-01222-t001:** Nash equilibria in joint probabilities.

Games	Classic Strategies	Quantum Strategies
Prisoner’s Dilemma	$\left(p_{1}^{N},q_{1}^{N}\right)$	$\left(w_{1}^{N},u_{1}^{N}\right)\;\vee \;\left(w_{2}^{N},u_{2}^{N}\right)$
Chicken Game	$\left(p_{2}^{N},q_{2}^{N}\right)\;\wedge \;\left(p_{3}^{N},q_{3}^{N}\right)$	$\left(w_{1}^{N},u_{1}^{N}\right)\;\vee \;\left(w_{2}^{N},u_{2}^{N}\right)$
Battle of the Sexes	$\left(p_{1}^{N},q_{1}^{N}\right)\;\wedge \;\left(p_{4}^{N},q_{4}^{N}\right)$	$\left(w_{3}^{N},u_{3}^{N}\right)$

**Table 2 entropy-25-01222-t002:** Restrictions on Nash equilibrium with noise.

Games	Classical Strategies	Quantum Strategies
Prisoner’s Dilemma	$0\leq \mu^{\left(DP\right)}\leq 1-\sqrt[{4}]{\frac{1}{3}}$	$0\leq \mu^{★(DP)}\leq 1-\frac{1}{\sqrt[{4}]{2}}$
Chicken Game	No restrictions	$0\leq \mu^{★(JP)}\leq \sqrt[{4}]{\frac{2}{3}}$
Battle of the Sexes	No restrictions	No restrictions

**Table 3 entropy-25-01222-t003:** Parameters of the Games with noise.

Games	Players	$\epsilon_{1}^{R}$	$\epsilon_{2}^{R}$	$\epsilon_{3}^{R}$	$\epsilon_{4}^{R}$	$\epsilon_{1}^{★R}$	$\epsilon_{2}^{★R}$	$\epsilon_{3}^{★R}$	$\epsilon_{4}^{★R}$
Prisoner’s Dilemma	Alice	1	0	1	1	1	0	1	1
	Bob	1	1	0	1	1	1	0	1
Chicken Game	Alice	1	1	1	0	1	1	1	0
	Bob	1	1	1	0	1	1	1	0
Battle of the Sexes	Alice	1	0	0	1	1	0	0	1
	Bob	1	0	0	1	1	0	0	1

**Table 4 entropy-25-01222-t004:** Probabilities of quantum games with noise.

Games	Players	$P^{R}$		$Q^{R}$		$P^{★R}$		$Q^{★R}$	
		$\epsilon_{1}^{R}$	$\epsilon_{2}^{R}$	$\epsilon_{1}^{R}$	$\epsilon_{3}^{R}$	$\epsilon_{1}^{★R}$	$\epsilon_{2}^{★R}$	$\epsilon_{1}^{★R}$	$\epsilon_{3}^{★R}$
Prisoner’s Dilemma	Alice	1	0	1	1	1	0	1	1
	Bob	1	1	1	0	1	1	1	0
Chicken Game	Alice	1	1	1	1	1	1	1	1
	Bob	1	1	1	1	1	1	1	1
Battle of the Sexes	Alice	1	0	1	0	1	0	1	0
	Bob	1	0	1	0	1	0	1	0

## Data Availability

No new data were created or analyzed in this study. Data sharing is not applicable to this article.

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
