# Peer review of "Joint Probabilities Approach to Quantum Games with Noise"

_entropy, 2023, doi:10.3390/e25081222_

Round 1
Reviewer 1 Report
preIn this work the Authors introduce a noise model for quantum games with joint probabilities. Then they utilize their approach to study the behavior of the typical games - prisoners dilemma, battle of sex and chicken. They study the behavior of Nash equlibria of these games. Next, they utilize their results to provide insight into the study of entanglement.
The results are sound and, in parts, interesting. I think the description of the model of the game and explicit formuals could moved to an appendix. In the current form it makes difficult to get to the most interesting part of the paper - regarding entanglement.
A have only two more technical comments:
1. Some of the equations do not fit in a single line. When moving thisngs around this should be fixed.
2. The resolution of Fig. 1 could be improved
Based on the above I recommend publiction after minor revisions.
Author Response
Reviewer 1
…preIn this work the Authors introduce a noise model for quantum games with joint probabilities. Then they utilize their approach to study the behavior of the typical games - prisoners dilemma, battle of sex and chicken. They study the behavior of Nash equlibria of these games. Next, they utilize their results to provide insight into the study of entanglement.
The results are sound and, in parts, interesting. I think the description of the model of the game and explicit formuals could moved to an appendix. In the current form it makes difficult to get to the most interesting part of the paper - regarding entanglement.
REFEREE 1
A have only two more technical comments:
- Some of the equations do not fit in a single line. When moving thisngs around this should be fixed.
RESPONSE
Thank you for pointing this out. We will move some more technical details to the appendix in order to make the paper more fluent and use a full-page format for the larger equations. The format of the journal is single column, so we should not have problems with any of the long equations.
REFEREE 1
2. The resolution of Fig. 1 could be improved
RESPONSE
We will improve the resolution of Figure 1.
REFEREE 1
Based on the above I recommend publication after minor revisions.
We thank the referee for his support of our work.
Reviewer 2 Report
The authors consider a quantum game with noise.
This type of game has been addressed by many authors previously and I do not see any crucial and original contributions to the game theory.
Besides, the author did follow recent advances in quantum game theory and did not cite even some related papers. For example I can find a lot of works on quantum games with noise, but none of them were mentioned in the article.
https://iopscience.iop.org/article/10.1088/1009-1963/16/4/008
https://journals.aps.org/pra/abstract/10.1103/PhysRevA.65.052320
Obviously, the authors are not the first guys working on the problem, but they are neglecting previous works and relations with them, which is a serious issue for drafting a scientific paper.
Besides, the "new correlation measure" presented has no mathematical or physical basis for discussing the quantum nature of the game. Non-classical nature can be measured by entanglement or quantum discord, and no other measure of quantum correlation is not known. There is no general proof or concrete evidence that their "measure" is truly a quantum correlation measure.
Furthermore, the authors do not discuss the developmental potential of their study, and their study is not of specific interest to the broader readership. The authors must need to mention any possible applications of their work in a more concrete way. For example,
Repeated games
https://link.springer.com/article/10.1007/s11128-021-03295-7
Extensive form games
https://link.springer.com/article/10.1007/s11128-022-03806-0
Phase transition
https://link.springer.com/article/10.1007/s11128-018-1918-6
And see more in review
https://link.springer.com/article/10.1007/s11128-018-2082-8
Due to the reasons above the article is not suitable for publication.
Author Response
Reviewer 2
The authors consider a quantum game with noise.
This type of game has been addressed by many authors previously and I do not see any crucial and original contributions to the game theory.
REFEREE 2
Besides, the author did follow recent advances in quantum game theory and did not cite even some related papers. For example I can find a lot of works on quantum games with noise, but none of them were mentioned in the article.
https://iopscience.iop.org/article/10.1088/1009-1963/16/4/008
https://journals.aps.org/pra/abstract/10.1103/PhysRevA.65.052320
Obviously, the authors are not the first guys working on the problem, but they are neglecting previous works and relations with them, which is a serious issue for drafting a scientific paper.
RESPONSE
We thank the referee for his comments, obviously, we are not the first to address the problem of noise in Quantum Games, and it was not our intention to convey this notion. Nevertheless, this paper builds on what we think is a novel concept in Quantum games: the “dissolution of dilemmas” due to entanglement. We cited this contribution in reference https://doi.org/10.1038/s41598-022-17072-8. So the main purpose of this work is to see how noise can disrupt the dissolution of dilemmas. We will make this point clearer in the new version of the manuscript.
We will also include mention to the suggested papers.
REFEREE 2
Besides, the "new correlation measure" presented has no mathematical or physical basis for discussing the quantum nature of the game. Non-classical nature can be measured by entanglement or quantum discord, and no other measure of quantum correlation is not known. There is no general proof or concrete evidence that their "measure" is truly a quantum correlation measure.
REPONSE
Our approach to the correlation measure operates on the joint probabilities that characterize the quantum or classical nature of the game for the proposed quantum game formalism and not on the traditional quantum correlation measures that operate on the system's quantum state. Likewise, the proposed correlation measure, like those traditional measures used in quantum information, is positive definite.
REFEREE 2
Furthermore, the authors do not discuss the developmental potential of their study, and their study is not of specific interest to the broader readership. The authors must need to mention any possible applications of their work in a more concrete way. For example,
Repeated games
https://link.springer.com/article/10.1007/s11128-021-03295-7
Extensive form games
https://link.springer.com/article/10.1007/s11128-022-03806-0
Phase transition
https://link.springer.com/article/10.1007/s11128-018-1918-6
And see more in review
https://link.springer.com/article/10.1007/s11128-018-2082-8
Due to the reasons above the article is not suitable for publication.
RESPONSE
We have included a discussion of the possible applications of our work to quantum channels in the discussion and conclusions of the paper.
In addition to the realm of noise quantum games, we have explored the potential of conceiving noise quantum games as functional information channels endowed with a specific channel capacity. This notion was initially hinted at in our prior work [19], wherein we introduced the concept of an information capacity for quantum games
from joint probabilities. Within this framework, quantum games from joint probabilities are cast as distinct information channels, with each individual game assuming the role of a unique channel.
In the context of [19], we have demonstrated that strategic approaches that effectively mitigate dilemmas in scenarios like the Battle of the Sexes exhibit a notably superior capacity compared to the Flip channel. Consequently, this novel approach may serve as a promising avenue toward the development of high-capacity information channels that exhibit resilience against noise. Importantly, the capacity of such a channel directly correlates with its ability to withstand the influence of noise and decoherence.
It is worth highlighting that the amalgamation of the formalism proposed for quantum games, incorporating noise from joint probabilities with the information capacity framework delineated in [19], presents a compelling and innovative avenue for advancing the transmission of information within the realm of quantum information channels.
Other applications for games with joint probabilities are related to infinitely repeated quantum games and the efficiency of these strategies with respect to their simple
counterpart, also showing the Quantum Folk Theorem [33]. Likewise, the basis for many important game theoretical models in economics are the games of extensive
form in their quantum version. A recent application of these games is the quantization of the Angel problem [34]. Another interesting application is related to single qubit games with graphical analysis, it is possible to notice that the curves that represent the gains of the players present a behavior similar to the curves that give rise to a phase transition in thermodynamics [35]. These transitions are
associated with optimal strategy changes and occur in the absence of entanglement and interaction between the players.
Finally, the quantum games with noise propose to relate to the mapping of game concepts onto statistical mechanics describing thermalization processes using classical game concepts [36]. Exploring the quantum game counterpart could be illuminating for identifying decoherence processes understood through quantum game equilibria as described here and [19].
With these comments, we have expanded the new discussion of the paper. We thank the referee for his comments and concerns.
Round 2
Reviewer 2 Report
Now the paper looks publishable.